# CBP/P300 Inhibition Impairs CD4+ T Cell Activation: Implications for Autoimmune Disorders

**DOI:** 10.3390/biomedicines12061344

**Published:** 2024-06-18

**Authors:** Lucas Wilhelmus Picavet, Anoushka A. K. Samat, Jorg Calis, Lotte Nijhuis, Rianne Scholman, Michal Mokry, David F. Tough, Rabinder K. Prinjha, Sebastiaan J. Vastert, Jorg van Loosdregt

**Affiliations:** 1Center for Translational Immunology, University Medical Center Utrecht, 3584 CX Utrecht, The Netherlands; l.w.picavet@umcutrecht.nl (L.W.P.); a.a.k.samat-3@umcutrecht.nl (A.A.K.S.); j.j.a.calis-2@umcutrecht.nl (J.C.); l.nijhuis@umcutrecht.nl (L.N.); r.c.scholman@umcutrecht.nl (R.S.); b.vastert@umcutrecht.nl (S.J.V.); 2Department of Experimental Cardiology, University Medical Center Utrecht, 3584 CX Utrecht, The Netherlands; m.mokry@umcutrecht.nl; 3Immunology Research Unit, Medicines Research Centre, GlaxoSmithKline, Stevenage SG1 2NY, UK; david.f.tough@gsk.com (D.F.T.); rabinder.prinjha@gsk.com (R.K.P.); 4Department of Pediatric Rheumatology and Immunology, University Medical Center Utrecht, 3584 CX Utrecht, The Netherlands

**Keywords:** P300/CBP, T cell activation, autoimmune diseases, JIA, H3K27ac, iCBP112, cytokines

## Abstract

T cell activation is critical for an effective immune response against pathogens. However, dysregulation contributes to the pathogenesis of autoimmune diseases, including Juvenile Idiopathic Arthritis (JIA). The molecular mechanisms underlying T cell activation are still incompletely understood. T cell activation promotes the acetylation of histone 3 at Lysine 27 (H3K27ac) at enhancer and promoter regions of proinflammatory cytokines, thereby increasing the expression of these genes which is essential for T cell function. Co-activators E1A binding protein P300 (P300) and CREB binding protein (CBP), collectively known as P300/CBP, are essential to facilitate H3K27 acetylation. Presently, the role of P300/CBP in human CD4+ T cells activation remains incompletely understood. To assess the function of P300/CBP in T cell activation and autoimmune disease, we utilized iCBP112, a selective inhibitor of P300/CBP, in T cells obtained from healthy controls and JIA patients. Treatment with iCBP112 suppressed T cell activation and cytokine signaling pathways, leading to reduced expression of many proinflammatory cytokines, including IL-2, IFN-γ, IL-4, and IL-17A. Moreover, P300/CBP inhibition in T cells derived from the inflamed synovium of JIA patients resulted in decreased expression of similar pathways and preferentially suppressed the expression of disease-associated genes. This study underscores the regulatory role of P300/CBP in regulating gene expression during T cell activation while offering potential insights into the pathogenesis of autoimmune diseases. Our findings indicate that P300/CBP inhibition could potentially be leveraged for the treatment of autoimmune diseases such as JIA in the future.

## 1. Introduction

The paralogous proteins P300 (E1A binding protein P300) and CBP (CREB binding protein), collectively referred to as P300/CBP, play a central role in the transcriptional regulation of gene expression [1,2,3]. P300/CBP facilitate transcription through epigenetic mechanisms, specifically via the acetylation of histone 3 Lysine 27 (H3K27ac), which is associated with active promoter and enhancer regions [4,5,6,7,8,9]. Moreover, they act as transcriptional co-activators by binding to acetylated histones using their bromodomain (BRD), thereby recruiting transcription factors (TFs) along with the transcriptional machinery [10,11].

Epigenetic mechanisms play a crucial role in T cell development, differentiation, and autoimmune diseases, as evidenced by numerous studies [12,13,14,15,16,17,18]. The process of T cell lineage commitment into Th1 and Th2 subtypes involves specific patterns of histone modifications and P300 distribution, particularly at the sites of cytokine genes [13,14,19,20,21]. Moreover, the regulation of genes specific to T cell lineages is governed by large clusters of transcriptional enhancers, also known as super-enhancers (SE) [22]. These SEs are highly enriched for P300/CBP, transcription factors, co-activators, and RNA polymerase II, indicating their vital role in gene expression. Importantly, these SEs are linked with genes encoding for cytokines and cytokine receptors and are enriched with lineage-defining transcription factors, such as T-BET, GATA3, and ROR-γt [23]. Moreover, the transcription factor NFAT, which is critical for the activation and differentiation of T cells, is modulated by P300/CBP [24]. This modulation involves P300/CBP enhancing NFAT’s transcriptional activity through the acetylation of histones, thereby promoting a chromatin structure that is more open and favorable for gene transcription. Mice with diminished CBP expression (CBP+/−) exhibited hematopoietic irregularities along with reduced numbers of differentiated B and T cells [25]. The targeted deletion of CBP or P300 in murine thymocytes leads to impaired T cell development and reduced peripheral T cell numbers [16]. Additionally, the presence of CBP is crucial for maintaining standard CD4/CD8 ratios and the formation of CD8+ effector and memory cells [15,16,26]. Despite the evidence linking CBP/P300 with the promoters of pro-inflammatory cytokines such as IL-2, IL-4, and IFN-γ, the precise role of P300/CBP in T cell activation and autoimmune diseases such as Juvenile Idiopathic Arthritis (JIA) remains poorly understood [12,13,14,20].

Juvenile Idiopathic Arthritis (JIA) encompasses a spectrum of chronic childhood arthritic conditions characterized by joint inflammation. Notably, T cells in JIA have been found to produce high levels of pro-inflammatory cytokines, primarily in the synovial fluid (SF) of inflamed joints, contributing to the disease’s pathology [27,28]. In oligoarticular JIA (oJIA), which is characterized by inflammation in fewer than five joints, more Th1 and Th17 cells are present in the SF of joints with arthritis, indicating a skewing towards type 1 immunity [29]. However, the exact mechanism behind the increased activation of the immune system remains elusive.

In this study, we explored the role of P300/CBP in T cell activation and its implications for autoimmune disorders, particularly JIA. Our investigation is aimed at understanding the molecular mechanisms underlying T cell activation and the potential impact of P300/CBP inhibition. By studying T cells from both healthy donors and patients with JIA, we aimed to uncover key pathways and gene expression changes influenced by P300/CBP activity. This research provides insights into the therapeutic potential of targeting P300/CBP in autoimmune diseases.

## 2. Materials and Methods

### 2.1. Cell Culture

PBMCs were isolated from five healthy controls using Ficoll Isopaque density gradient centrifugation. SFMCs were similarly isolated from the synovial fluid of five oligoarticular JIA patients with active disease at the time of sampling. CD4-positive T cells were isolated using the microbead CD4 isolation kit (Miltenyi Biotec, Bergisch Gladbach, Germany) using negative selection on the autoMACS. Cells were cultured in RPMI 1640 culture medium (Thermo Fisher, Waltham, MA, USA) supplemented with 10% human serum (Merck Life Science, Darmstadt, Germany), 1% penicillin-streptomycin (Merck Life Science), and 1% glutamine (Merck Life Science). Cells were pre-incubated for 1 h with 6 μM iCBP112 (GlaxoSmithKline; GSK, Brentford, London, UK) or an equivalent volume of DMSO prior to activation with human T-activator CD3/CD28 dynabeads (1 cell, 3 beads; Thermo Fisher) and cultured for 16 h in the presence of 6 uM iCBP112 at 37 degrees and 5% CO_2_. The human biological samples were sourced ethically, and their research use was in accordance with the terms of the informed consent under an IRB/EC-approved protocol.

### 2.2. RNA-Sequencing

Total RNA was extracted using the RNAeasy kit (Qiagen, Hilden, Germany) according to the manufacturer’s protocol. mRNA was isolated using NEXTflex Poly(A) Beads (Bio Scientific, Austin, TX, USA), and libraries were prepared using NEXTflex Rapid Directional RNA-seq Kit (Bio Scientific). The quality of the RNA was assessed by the bioanalyzer on a pico-chip. Samples were sequenced using 75 bp single-end reads on an Illumina Nextseq 500 platform (Illumina Inc., San Diego, CA, USA) through the Utrecht sequencing facility (USEQ, Utrecht, The Netherlands). Reads aligned to the human genome (version GRCh37) and transcriptome (ENSEMBL version 37.74) with STAR version 2.4.2a were quantified at the gene level. Sample quality was evaluated as the number of expressed genes, and by Principal Component Analysis (PCA). Samples with fewer than 11,000 expressed genes were discarded. Samples were analyzed for differential gene expression with Voom-Limma using the eBayes functionality [30]. Gene set expression analysis was performed using CAMERA [31] with gene set information from MSigDB (version 6.2) [32]. In each case, TMM normalized expression values were modeled using a linear model that included group inhibitor treatment, disease status information, and donor information. Sample groups that differed by inhibitor treatment or by disease status were contrasted to examine differential expression. Differential expression statistics were corrected for multiple testing by the Benjamini-Hochberg method [33]. Venn diagrams of differentially expressed genes were made using the Biovenn web application [34]. Functional enrichment of differentially expressed gene lists was detected using Toppfun Suite [35]. Gene set enrichment analysis (GSEA) was conducted using the GSEA software 2.0 released by Broad Institute [32,36].

### 2.3. shRNA-Induced Knockdown of CREBBP in Jurkat Cells

HEK293T cells were cultured O/N at 37 °C, 5% CO_2_ in Dulbecco’s Modified Eagle Medium with GlutaMax (Gibco), supplemented with 10% heat-inactivated fetal calf serum (Sigma-Aldrich, St. Louis, MO, USA) and 100 U/mL penicillin and 100 mg/mL streptomycin (Gibco) to confluency and transfected with 5 μg target plasmid DNA (Sigma Aldrich, MISSION^®^ lentiviral shRNA control or shCREBBP; TRCN0000011027), 3.4 μg pPAX2 (Addgene, Watertown, MA, USA, #35002) and 1.6 μg pMD2G (Addgene, #12259) using polyethyleneimine (PEI MAX, Polysciences Europe, 24765-1) to produce viral particles. Post 24 h incubation, the virus-containing medium was filtered and utilized to transduce wildtype Jurkat cells (Clone E6.1; ATCC) using 5 µg/mL polybrene (Santa Cruz Biotechnology, Dallas, TX, USA) overnight. After 24 h, a second viral hit was performed by repeating the previous step. Transduced cells were cultured in RPMI 1640 medium (Gibco) supplemented with 1% L-glutamine (Merck Life Science), 10% heat-inactivated fetal bovine serum (Sigma-Aldrich, St. Louis, MO, USA) and 100 U/mL penicillin, and 100 mg/mL streptomycin (Gibco) at 37 °C and 5% CO_2_. After 48 h, cells were selected for successful transduction using 10 ug/mL puromycin (Merck Life Science, Darmstadt, Germany). Subsequently, cells were activated for 4 h with PMA (20 ng/mL; Sigma-Aldrich) and ionomycin (1 μg/mL; Calbiochem, Darmstadt, Germany) prior to qPCR analysis.

### 2.4. qPCR Analysis

Cells were lysed in RLT buffer (RNeasy kit, Qiagen), and RNA was isolated using the manufacturer’s protocol. cDNA synthesis was performed using the iScript cDNA synthesis kit (Bio-Rad, Hercules, CA, USA). qPCR analysis was performed on the Quantstudio 3 real-time PCR system with SYBR select master mix (Thermo Fisher, Waltham, Massachusetts, USA) and the following primer sets: B2M forward primer TGCTGTCTCCATGTTTGATGTATCT, B2M reverse primer TCTCTGCTCCCCACCTCTAAGT; IL2 forward primer AACTCACCAGGATGCTCACATTTA, IL2 reverse primer TCCCTGGGTCTTAAGTGAAAGTTT; IFNγ forward primer GCAGAGCCAAATTGTCTCCT, IFNγ reverse primer ATGCTCTTCGACCTCGAAAC; IL4 forward primer TGCCGGCAACTTTGTCCACGG, IL4 reverse primer GTCTGTTACGGTCAACTCGGTGCA; IL17α forward primer CCGTGGGCTGCACCTGTGTC, and IL17α reverse primer GGGAGTGTGGGCTCCCCAGA.

### 2.5. Luminex Assay

The supernatant was collected at the end of activation and stored at −80 °C before protein analysis. The Luminex multiplex assay was performed using antibodies against IL2, IFNγ, IL4, and IL17α as described previously [37].

### 2.6. Analysis

Data were analyzed using GraphPad Prism 10.2.1, and group comparisons were tested for significance using paired *t* tests.

## 3. Results

### 3.1. P300/CBP Regulate Pro-Inflammatory Pathways Associated with T Cell Activation

To assess the role of P300/CBP in T cell activation, primary human CD4+ T cells from five different donors were treated with the P300/CBP-specific inhibitor iCBP112 for 1 h prior to stimulation using anti-CD3/CD28 coated beads for 16 h. Next-generation RNA sequencing was performed on these cells to evaluate the transcriptome-wide expression of mRNA. Treatment with the P300/CBP inhibitor resulted in 2562 significantly (FDR < 0.05) differentially expressed genes (DEs), of which 1276 genes were downregulated following treatment with iCBP112 (Figure 1A,B). Many of the downregulated genes are associated with immune regulation, including interleukins, chemokines, and CD markers (Figure 1C). More specifically, pathway analysis revealed that P300/CBP inhibition impaired the expression of genes associated with processes involved in cytokine signaling (Figure 1D–F). These findings indicate that the inhibition of P300/CBP results in reduced T cell activation, as indicated by impaired cytokine production.

### 3.2. P300/CBP Inhibition Preferentially Inhibits the Expression of Proinflammatory Cytokines

To gain further insights into the regulatory role of P300/CBP in proinflammatory cytokine expression, we assessed the impact of iCBP112 on key proinflammatory T cell cytokines, including IL2, IL4, IFNγ, and IL17α, since a noticeable decrease in the expression of each of these cytokines was observed in our RNA-seq data (Figure 2A). These findings were validated by qPCR analysis performed on independent donors (Figure 2B). Moreover, the quantities of each of these cytokines present in the supernatants of iCBP112-treated T cells were also reduced compared to control cells (Figure 2C). In conclusion, these data demonstrate that iCBP112 inhibits the expression of proinflammatory cytokines after T cell activation.

### 3.3. P300/CBP Inhibition Down-Regulates Genes Associated with Enhanced Expression in T Cells from Patients with Juvenile Idiopathic Arthritis

As we observed that P300/CBP inhibition suppresses CD4+ T cell activation and proinflammatory cytokine expression, we hypothesized that the inhibition of P300/CBP using iCBP112 could help to restore T cell homeostasis in autoimmunity. Here, we focused on Juvenile Idiopathic Arthritis (JIA) since we have already demonstrated in the past that T cells from the inflamed joints of these patients have an activated phenotype and display increased H3K27Ac of enhancers and promoters associated with genes involved in T cell activation [18].

CD4+ T cells were isolated from the blood of five healthy controls or the synovial fluid (SF) from inflamed knee joints of five JIA patients, activated with anti-CD3/CD28 for 16 h, and mRNA expression in these cells was compared by RNA-seq. We identified numerous transcriptional changes (7623 genes, FDR < 0.05) between healthy and patient-derived cells. JIA-derived CD4+ cells showed increased expression of genes related to inflammation, including many interleukins, chemokines, and CD markers involved in immune system signaling (Figure 3C), which were associated with cytokine-related biological processes (Figure 3B). These data indicate an enhanced state of activation in T cells derived from JIA SF compared to controls.

The administration of iCBP112 one hour prior to the 16 h activation of CD4+ T cells from JIA patients resulted in significant transcriptional changes with 6189 DE genes (FDR < 0.05), of which 3267 genes were significantly downregulated after treatment (Figure 3D–F and Figure A1A). Consistent with what was observed in healthy control cells, many of the DE genes were associated with pathways related to cytokine signaling (Figure 3E). Notably, we found that pathways that were increased in cells from JIA compared to healthy controls were inhibited by iCBP112 treatment, implying that P300/CBP inhibition preferentially reduces the expression of JIA-associated genes. To confirm this, we performed gene set enrichment for genes inhibited by iCBP112 (foldchange < 0.5, FDR < 0.05) in the dataset comparing JIA with healthy control cells (Figure 3G). Indeed, we identified that genes inhibited by iCBP112 were significantly enriched in the set of genes that are upregulated in JIA. Moreover, out of all the genes significantly inhibited by iCBP112, 429 genes were significantly upregulated in JIA and were notably associated with cytokine signaling pathways, among others (Figure 3H). Further analysis revealed that the treatment of JIA cells with iCBP112 resulted in a significant decrease in the expression of proinflammatory cytokines IL2, IL4, IFNγ, and IL17α (Figure 3I and Figure A1B). In summary, treatment of JIA patient-derived CD4+ T cells with iCBP112 induced transcriptional changes, downregulated immune-related pathways, and reduced the expression of JIA-associated genes. These findings suggest the therapeutic potential of iCBP112 for JIA.

## 4. Discussion

In this study, we have elucidated the central role of P300/CBP in modulating T cell activation and cytokine expression. By utilizing next-generation RNA-sequencing analysis, we observed significant changes in the gene expression profiles of CD4+ T cells subjected to treatment with iCBP112, a P300/CBP-specific inhibitor. This intervention led to the suppression of key immune mediators, including interleukins, chemokines, and CD markers, which are crucial for T cell activation and orchestration of the immune response. In addition, when applying P300/CBP inhibition in CD4+ T cells extracted from the site of inflammation in JIA patients, we identified a preferential reduction in the expression of genes associated with JIA. Further analysis revealed that genes downregulated by iCBP112 treatment are predominantly associated with cytokine signaling pathways. This highlights the critical role of P300/CBP in cytokine regulation during T cell activation. Our results underscore the involvement of P300/CBP in shaping the immune response, highlighting its potential as a therapeutic target for auto-inflammatory and autoimmune diseases.

Recent studies have explored the therapeutic applications of CBP/P300-specific inhibitors in both cancer and autoimmune diseases, demonstrating their promise [38,39]. Specifically, inhibitors targeting the bromodomain of P300/CBP have been shown to hinder Treg cell differentiation and function, offering a strategy to enhance effector responses in the context of cancer [40]. CBP30, a selective inhibitor of the CBP/P300 bromodomain, exhibits anti-inflammatory effects across various primary cells and demonstrates CBP/P300 is critical for IL17α signaling [41]. Our study leverages iCBP112, which similarly targets the P300/CBP bromodomain, reinforcing the idea that selective inhibition of this domain can suppress T cell activation and reduce pro-inflammatory cytokine production, thereby presenting new avenues for targeted treatment in auto-inflammatory and autoimmune conditions.

We analyzed P300 and CBP expression using RNA sequencing data and found no significant differences in their levels between healthy controls, JIA patients, and following iCBP112 treatment. These findings indicate that the effects of P300/CBP inhibition and JIA pathogenesis are likely due to changes in protein activity rather than expression levels. The BRD domain is essential for mediating interactions between transcription factors and DNA, and it is possible that iCBP112 is disturbing these interactions [42].

It is important to note that our study focuses on CD4-positive T cells, and we did not differentiate between memory and naïve T cell subsets. Despite this limitation, our findings provide valuable insights into T cell responses. Further research is needed to elucidate the involvement of specific T cell subsets in our experimental context.

Treatment with iCP112 specifically reduced the expression of disease-associated genes in T cells from JIA patients. This aligns with our prior research, which identified disease-specific enhancer profiles in JIA T cells and showed that targeting these enhancers with pan BET inhibitors reduces the expression of both disease-associated and pro-inflammatory genes [18]. Additionally, both histone acetyltransferase (HAT) and bromodomain inhibitors of P300/CBP have proven effective in mitigating the inflammatory profiles of synovial fibroblasts derived from rheumatoid arthritis patients [39]. Further research is required to explore the therapeutic efficacy of P300/CBP-specific HAT inhibition in JIA and to extend these findings to other autoimmune diseases.

In conclusion, our study highlights the central role of P300/CBP regulating transcription, T cell activation and cytokine signaling. The inhibition of P300/CBP, particularly within inflammation sites, preferentially suppresses the expression of JIA-associated genes, pointing to the potential of P300/CBP inhibition as a therapeutic strategy for autoimmune diseases such as JIA. Future research should focus on unraveling the full therapeutic implications of specific P300/CBP- inhibition in JIA in the overall context of other reported effects in the immune system and assess its applicability to a wider spectrum of autoimmune disorders.

## Figures and Tables

**Figure 1 biomedicines-12-01344-f001:**
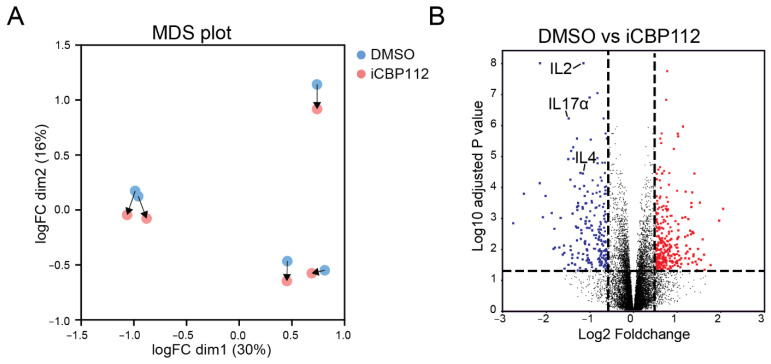
P300/CBP regulates transcriptional pathways involved in general T cell activation. RNA sequencing was performed on CD4-positive T cells isolated from the blood of five healthy donors and subsequently treated with iCBP112 before stimulation with aCD3/CD28 beads for 16 h. (**A**) Principal component analysis comparing CD4 T cells isolated from healthy control blood following treatment with iCBP112. (**B**) Volcano plot of the comparison between healthy control cells treated with iCBP112 and those treated with DMSO control. Genes significantly downregulated after treatment with iCBP112 are represented by blue dots, red dots depict genes that are significantly upregulated. (**C**) Selection of genes associated with immune activation exhibiting downregulated expression after iCBP112 treatment. (**D**) Gene Ontology (GO) terms representing pathways associated with genes showing differential expression after iCBP112 treatment. (**E**) Gene set enrichment analysis (GSEA) of Biocarta cytokine pathway, KEGG cytokine receptor interaction, and WP overview of proinflammatory and profibrotic mediator’s gene sets in the differentially expressed dataset following iCBP112 treatment. Colors demonstrate foldchange after iCBP112 treatment on a scale from red (high) to blue (low). (**F**) Heat map depicting genes within the Biocarta cytokine pathway. Colors indicate relative RNA expression on a scale from red (high) to blue (low).

**Figure 2 biomedicines-12-01344-f002:**
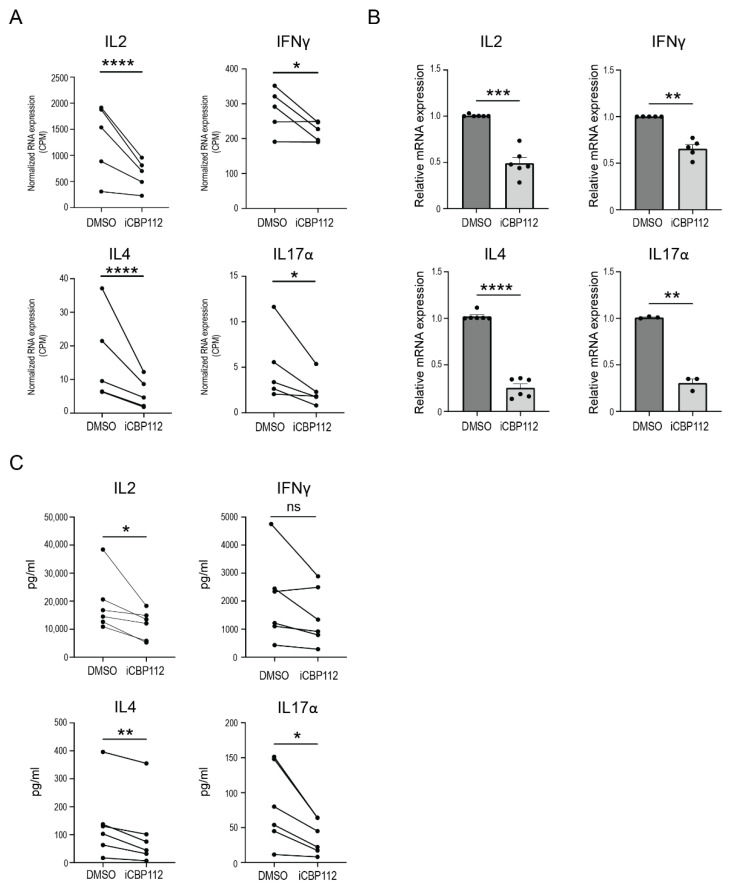
iCBP112-mediated inhibition of key proinflammatory cytokine expression. (**A**) Normalized RNA expression (counts per million, CPM) of IL2, IFNγ, IL4, and IL17α genes extracted from the RNA-sequencing dataset following iCBP112 treatment. (**B**) Relative mRNA expression of IL2, IFNγ, IL4, and IL17α as measured by quantitative polymerase chain reaction (qPCR). (**C**) Quantitative analysis of IL-2, IFN-γ, IL-4 and IL-17α protein expression measured by Luminex. *p*-values were calculated using a paired *t*-test, n = 6, ns: not significant, *: *p* < 0.05, **: *p* < 0.01, ***: *p* < 0.005, ****: *p* < 0.001.

**Figure 3 biomedicines-12-01344-f003:**
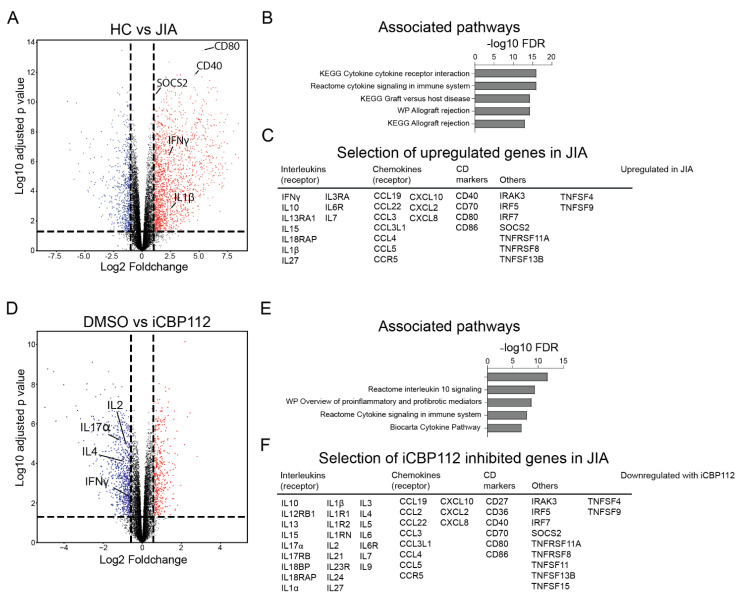
iCBP112 preferentially inhibits JIA-associated genes in synovial T cells. CD4+ T cells were isolated from the synovial fluid of five JIA patients or peripheral blood of five healthy controls and subsequently treated with iCBP112 before stimulation with aCD3/CD28 beads for 16 h. mRNA expression was compared by RNA sequencing. (**A**) Volcano plot illustrating the comparison between DMSO-treated healthy control cells and JIA-derived cells. Genes significantly downregulated in JIA are represented by blue dots, red dots depict genes that are significantly upregulated. (**B**) GO-terms representing the pathways associated with genes, showing differential expression in JIA. (**C**) Selected genes associated with immune regulation that are differentially expressed, both upregulated and downregulated in JIA. (**D**) Volcano plot illustrating the comparison between JIA T cells treated with iCBP112 and those treated with DMSO control. Genes significantly downregulated after iCBP112 treatment are represented by blue dots, red dots depict genes that are significantly upregulated. (**E**) GO-terms representing the pathways associated with genes showing differential expression after iCBP112 treatment in JIA T cells. (**F**) A selection of genes associated with immune regulation that have downregulated expression after iCBP112 treatment in JIA T cells. (**G**) Gene set enrichment analysis (GSEA) depicting iCBP112-inhibited genes in the differentially expressed dataset of HC versus JIA T cells. Colors demonstrate foldchange in JIA T cells on a scale from red (high) to blue (low). (**H**) Venn diagram of iCBP112-inhibited genes in JIA and genes that are upregulated in JIA. Go-terms are shown for genes that are present in both gene lists. (**I**) Normalized expression (CPM) of IL2, IFNγ, IL4, and IL17α genes in JIA T cells treated with iCBP112. *p*-values were calculated using a paired *t*-test, n = 5 ***: *p* < 0.005, ****: *p* < 0.001.

## Data Availability

The original contributions presented in the study are included in the article, further inquiries can be directed to the corresponding author.

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
