# Peer review of "CBP/P300 Inhibition Impairs CD4+ T Cell Activation: Implications for Autoimmune Disorders"

_biomedicines, 2024, doi:10.3390/biomedicines12061344_

Round 1
Reviewer 1 Report
Comments and Suggestions for Authors
The study investigates the effect of CBP/P300 inhibition on T cell activation in terms of the production of pro-inflammatory cytokines in Juvenile Idiopathic Arthritis. Some comments are listed below:
(1) It is preferred to see those selected DEGs highlighted in each volcano plot. There is no need to highlight all of them but some like IL2, IFNg, IL4 and IL17A.
(2) Do you think disrupting the interaction between CBP and CREB or inhibiting CREB activity will have similar results as what you observed with i-CBP112? Do you observe any CREB target genes like NR4A2, STC1, PDE4B etc in the downregulated DEGs upon i-CBP112 treatment?
(3) For the tables of selected differential expressed genes like 1C, 3C and 3F, it would be better to specify whether they are down-regulated or up-regulated genes. Besides, the sample size (n = ?) for RNAseq data should be specified in the figure as well.
(4) As SFMCs from JIA patients are used in this study, it is preferred to have a brief introduction about JIA disease in the beginning.
Reviewer 2 Report
Comments and Suggestions for Authors
The manuscript experimentally investigated the regulatory role of P300/CBP in regulating gene expression during T cell activation. The approach is fundamental, and some nice experimental data is provided. Here are some comments for further clarification/consideration:
1) Please quantify the level of P300/CBP in T cells derived from both healthy controls and JIA patients. This comparison may be more directly related to the function of P300/CBP in regulating gene expression during T cell activation.
2) The gene expression before and after the inhibition of P300/CBP should be provided to directly evaluate the function of P300/CBP.
3) Is there a way to conduct Cas-9 knockout of P300/CBP? How to make sure P300/CBP is really inhibited?
4) The last paragraph in the Introduction section should belong to the Results. The authors do not need to talk about the results/findings in the Introduction section. Instead, they can provide an overview/outline of the problem they investigated in the last paragraph.
5) Please show the time for the data collection. Are those T cells memory or naïve T cells? The subtypes are important. Discussions are needed for this.
Comments on the Quality of English LanguageSome typos are found. Generally good writing.
Round 2
Reviewer 2 Report
Comments and Suggestions for Authors
The authors have addressed my comments. Thanks.
Comments on the Quality of English LanguagePlease double check the manuscript for typos.